# From State Spaces to Semigroups: Leveraging Algebraic Formalism for Automated Planning

**Alice Petrov, Christian Muise**

Queen's University, Canada
17ap87@queensu.ca, christian.muise@queensu.ca

## Abstract

This paper introduces an algebraic formalism linking transformation semigroups and the state transition systems induced by classical planning problems. We investigate some basic planning problems with interesting properties and establish fundamental characteristics of the corresponding semigroups, such as their ideals and Green's relations. Furthermore, we leverage semigroup theory to propose new approaches to existing concepts in automated planning, including the identification of landmark actions and the study of dead ends. We demonstrate that algebraic results can be applied to facilitate an understanding of a planning problem's state space and explore its solutions, thus verifying the relevance and effectiveness of such formal modeling.

## 1 Introduction

According to the majority of analysts, Moore's law will come to an end by 2025 (Waldrop 2016). To meet growing technological demands, rather than relying on the exponential growth of computational power, we must restrict resources and analyze what can be computed within those limitations. Thus, it is important to study the solution spaces of classical planning problems.

A semigroup is a closed, associative algebraic structure; anything with states, inputs, and outputs can be studied in semigroup terms. Semigroups are a useful framework for understanding complex systems because they allow us to identify substructures and their interactions. In fact, semigroups have been argued to be the underlying mathematical structures of computers (Egri-Nagy 2017), and they have been used to prove a number of classical results related to automata theory (Colcombet 2011). However, despite its close correspondence to automata theory, this application has yet to be extended to the transition systems induced by automated planning.

In this work we develop an algebraic model of classical planning problems, where a transformation semigroup can be used to represent actions and sequences of actions. The binary operation of the semigroup represents action composition, where, when viewed as a transformation, the output of one action is the input to another. Using semigroups to model planning problems provides several benefits, including:

1. Formalism: Semigroups provide a formal mathematical structure that can be used to reason about planning problems and develop algorithms for solving them.
2. Abstraction: Semigroups allow for the abstraction of specific details of a planning problem, which can make it easier to develop general-purpose algorithms that can be applied to a wide range of problems.

With respect to computer science systems, algebraic techniques have already been applied to the formal study of state transition systems in other areas (Letichevsky 2005; Cain 2009). Using semigroup theory takes mathematical effort. However, the results are easily applied. Low-level optimizations benefit from knowing the fewest number of states needed to perform an algorithm, and mathematical formalism can turn the problem of asking for all conceivable solutions, rather than a single solution, into a well-defined combinatorial question (Egri-Nagy 2017). The payoff could be that we find solutions we had never thought of.

When studying finite structures, such as the solution space induced by a planning problem, it is often beneficial to generate small instances, which can then be used to create new hypotheses and disprove conjectures. The size of the instances we can explore is determined by the available processing power and latest mathematical techniques. However, the basic presumption is that one can see well enough within the given limitations to enable extrapolation using theoretical reasoning, i.e., there is sufficient observational data to develop reliable theories (East et al. 2015). Semigroups are particularly amenable to studying solution spaces of small planning problems, and structural analysis can provide insight into bigger problems for which computing the entire solution space is not feasible.

## 2 Background

### 2.1 Classical Planning

The model underlying classical planning is typically described as

$$\Pi = \langle S, s_0, S_G, A, f \rangle$$

Where

- $S$ is a finite and discrete set of states
- $s_0 \in S$ is the initial state

- $S_G \subseteq S$ is the nonempty set of goal states
- $A$ is a set of actions
- $A(s) \subseteq A$ represents the set of actions applicable in state $s \in S$
- $f(a,s) = s'$ is the deterministic state transition function, which maps state $s$ to state $s'$ for $a \in A(s)$

A classical planning model $\Pi$ induces a directed graph $G$, which is called the state transition graph. From this point of view, the structure of a planning problem has two important mathematical properties:

1. *Closure*: The set of states $S$ is closed under the set of actions A, which means $\forall s \in S$ and $a \in A(s)$ we have $f(a,s) \in S$.
2. *Associativity*: Actions are associative, which means if the actions $a_1, a_2, a_3 \in A$ can be performed sequentially we have $(a_1 \circ a_2) \circ a_3 = a_1 \circ (a_2 \circ a_3)$

In an intuitive sense, these are closed systems that respect time and thus can be modeled as a semigroup, which is an algebraic structure that is closed and whose operation is associative.

## 2.2 Semigroups

Let $\Sigma$ be a set and $\circ$ be a binary operation, where a binary operation $\circ$ on a set $\Sigma$ is a map $\circ : \Sigma \times \Sigma \to \Sigma$. Formally, a semigroup is an algebraic structure $(\Sigma, \circ)$ which satisfies the following properties:

1. $Closure : \forall a, b, \in \Sigma : a \circ b \in \Sigma$
2. $Associativity : \forall a, b, c \in \Sigma : a \circ (b \circ c) = (a \circ b) \circ c$

A transformation semigroup consists of a set of states $S$ and a semigroup of transformations $(\Sigma, \circ)$, where $\Sigma$ is a set of functions from $S$ to $S$ and the binary operation $\circ$ is the concatenation of these functions. A transformation semigroup can be thought of as a generalization of permutation groups (which demand reversible transformations) (Morris et al. 2013).

In a set $\Sigma$, if there exists an element $e$ such that $ex = xe = x$ hold for all $x \in \Sigma$, then $e$ is called an identity. A semigroup that includes an identity element is referred to as a monoid. For a semigroup $\Sigma$, we define

$$\Sigma^1 = \begin{cases} \Sigma & \text{if } \Sigma \text{ has an identity} \\ \Sigma \cup \{1\} & \text{otherwise} \end{cases}$$

where 1 is the identity. We call $\Sigma^1$ the monoid obtained by adjoining an identity to $\Sigma$ if necessary (Cain 2013).

## 2.3 Instantiating a Planning Problem as a Transformation Semigroup

A transformation semigroup associated with a planning problem is composed of a set of states $S$ (which is identical to the set of states in a classical planning model) and a semigroup of transformations $(\Sigma, \circ)$. Here, $\Sigma$ represents all possible finite sequences of actions in $A$ (known as traces), and $\circ$ denotes the concatenation operation.

A generating set of a semigroup is a subset of the semigroup set such that every element of the semigroup can be expressed as a combination (under the semigroup operation) of finitely many elements of the subset. In the context of planning problems, the generating set is $A$, since any trace can be expressed as a combination of actions in $A$.

For each action $a \in A$, there is a corresponding transformation $\sigma_a$ in the set $\Sigma$. To complete an automaton, it is common practice to add a sink state and direct all previously undefined transitions to this new state (D'Angeli, Rodaro, and Wächter 2020). Similarly, to account for partial transformations, we introduce a sink state, denoted $s_0$. This effectively adds a zero element to the semigroup. Thus, the transformation $\sigma_a$ can be defined as follows:

$$\sigma_a(s) = \begin{cases} f(a,s) & a \in A(s) \\ s_0 & otherwise \end{cases}$$

An example of such a transformation can be found in the following section.

The subset of transformations $\{\sigma_a \mid a \in A\} \subseteq \Sigma$ forms the generating set for the entire semigroup $(\Sigma, \circ)$. Thus, the elements of $\Sigma$ are precisely the transformations which correspond to all of the possible traces induced by the planning problem.

In this abstract framework, semigroups provide a means of studying the solution space of a planning problem, where the solution space consists of traces. This allows us to investigate how traces are related and how they interact.

While semigroups offer an abstract perspective on planning problems, they do not inherently distinguish a start state or a set of goal states. Therefore, it is possible to introduce "start" and "end" actions to identify them explicitly.

# 3 Running Examples

## 3.1 Drive Domain

As our first running example, we use the very simple atomic transition system defined in "Graph-Based Factorization of Classical Planning Problems" (Wehrle, Sievers, and Helmert 2016). In this domain, a truck is supposed to drive from the initial location 1 to the goal location 4, where the goal location can be reached via intermediate locations 2 or 3. After adding "start", "end", and "skip" (do nothing) actions, as well as a sink state, we have the state transition graph seen in Figure 1 (note that we have omitted the self loop at each node in the graph induced by the "skip" action for clarity).

We define our set of states as

$$\{start, at\text{-}1, at\text{-}2, at\text{-}3, at\text{-}4, goal, sink\}$$

and label them with numbers from 1 to 7. We define our set of actions as

A = {begin, drive-1-2, drive-1-3, drive-2-4, drive-3-4, end, skip}

Let us now redefine this state transition system as a transformation semigroup $(\Sigma, \circ)$. Our set of states $S$ corresponds to those defined in the planning model, numbered 1 to 7 and illustrated in Figure 1. Our set of transformations $\Sigma$ is generated by the set of transformations corresponding to the set of actions $A$, listed in Table 1. In total, $\Sigma$ consists of 16 transformations, corresponding to all possible traces in this domain. As a more detailed example, let us take the action "drive-3-4" and illustrate the corresponding transformation

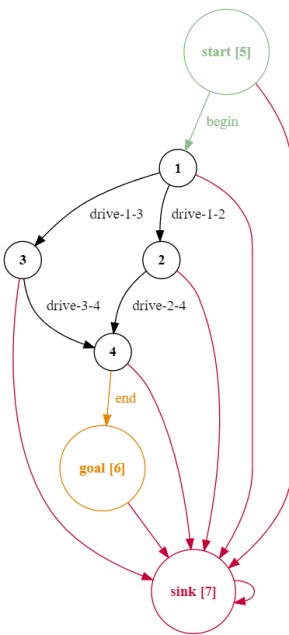

Figure 1: "Drive" state transition graph

| Action | Transformation |
|--------|----------------|
| begin | $\begin{bmatrix} 1 & 2 & 3 & 4 & 5 & 6 & 7 \\ 7 & 7 & 7 & 7 & 1 & 7 & 7 \end{bmatrix}$ |
| drive-1-2 | $\begin{bmatrix} 1 & 2 & 3 & 4 & 5 & 6 & 7 \\ 2 & 7 & 7 & 7 & 7 & 7 & 7 \end{bmatrix}$ |
| drive-1-3 | $\begin{bmatrix} 1 & 2 & 3 & 4 & 5 & 6 & 7 \\ 3 & 7 & 7 & 7 & 7 & 7 & 7 \end{bmatrix}$ |
| drive-2-4 | $\begin{bmatrix} 1 & 2 & 3 & 4 & 5 & 6 & 7 \\ 7 & 4 & 7 & 7 & 7 & 7 & 7 \end{bmatrix}$ |
| drive-3-4 | $\begin{bmatrix} 1 & 2 & 3 & 4 & 5 & 6 & 7 \\ 7 & 7 & 4 & 7 & 7 & 7 & 7 \end{bmatrix}$ |
| end | $\begin{bmatrix} 1 & 2 & 3 & 4 & 5 & 6 & 7 \\ 7 & 7 & 7 & 6 & 7 & 7 & 7 \end{bmatrix}$ |
| skip | $\begin{bmatrix} 1 & 2 & 3 & 4 & 5 & 6 & 7 \\ 1 & 2 & 3 & 4 & 5 & 6 & 7 \end{bmatrix}$ |

Table 1: "Drive" generators

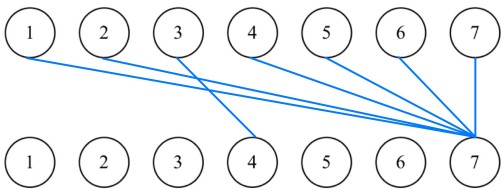

Figure 2: The transformation corresponding to the action "drive-3-4"

$[7, 7, 4, 7, 7, 7, 7]$, seen in Figure 2. Here, we see that the action "drive-3-4" sends state 3 to state 4, and is not applicable in any other state. Thus states 1, 2, 4, 5, 6, and 7 are sent to state 7 (the sink state).

## 3.2 Elevators Domain

Our next running example is the classic elevators domain. The situation is as follows: There are $n + 1$ floors in a building numbered from $0$ to $n$. For a given number of passengers, their current location (i.e., the floor they are on) and destination are specified. The planning problem is to devise a strategy for transporting passengers to their destinations. As a simple instance of this domain, we have one passenger $p_0$ and two floors, $f_0$ and $f_1$. In our initial state, our passenger is at floor $f_1$ and wants to go to floor $f_0$, and our lift is at floor $f_0$. We label our states from 1 to 14 and define our set of actions as follows.

A = {begin, up($f_0$, $f_1$), depart($f_0$, $p_0$), board($f_1$, $p_0$), down($f_1$, $f_0$), end, skip}

We redefine this transition system in the same manner as above. For example, after relabeling states with numbers, the grounded action up($f_0$, $f_1$) corresponds to the transformation

$$[2, 14, 14, 3, 6, 14, 14, 7, 14, 14, 14, 14, 14, 14]$$

Thus we have that the action "up($f_0$, $f_1$)" sends state 1 to state 2, state 4 to state 3, state 5 to state 7, and state 8 to state 7. The action is inapplicable in the remaining states, and therefore sends them to state 14 (the sink state).

## 4 Landmark Heuristics

Action landmarks are a specific type of landmark that represent key actions or events that must occur in order to achieve a goal (Karpas and Domshlak 2009). Once identified, action landmarks can be used in combination with other landmark-based heuristics, such as critical path and additive heuristics, to find an optimal solution (Helmert and Domshlak 2009).

A semigroup can be used to decide if a goal state in a planning problem is reachable by examining the structure of the semigroup and checking whether the goal state belongs to the set of reachable states. The set of reachable states can be generated by applying the semigroup operation (action concatenation) repeatedly, starting from the initial state and applying all possible sequences of actions. This reachability problem has been addressed in semigroup literature. More specifically, the *Cayley semigroup membership problem* is a decision problem that asks whether a given element belongs to a particular semigroup. Given a finite semigroup $\Sigma$ and an element $x$ of $\Sigma$, the problem is to determine whether $x$ can be expressed as a product of elements from a subset of $\Sigma$, where the product is taken using the semigroup operation (Fleischer 2022). The Cayley semigroup membership problem is NL-complete in general for finite semigroups (Fleischer 2018a). For groups, the problem can be solved in deterministic log-space (Fleischer 2018b).

In the case of automated planning, we are most interested in finding a transformation from the initial state to a goal state, as the existence of such an element is equivalent to

solvability in the problem we are modelling. In order to test whether a given action is a landmark, we can remove the generator corresponding to the action of interest and test if a transformation from the initial state to a goal state is still an element of the semigroup. If a transformation from the initial state to the goal state is no longer present, we have identified an action landmark. This demonstrates that there exist links between theoretical concepts in semigroup theory and automated planning. Furthermore, by inspecting ideals and $\mathcal{D}$-classes, semigroups can not only identify action landmarks, but also tell us about the structure of an action landmark and how its removal affects our solution space globally. We define ideals in the following section, and discuss $\mathcal{D}$-classes in Section 6.

## 4.1 Ideals

Intuitively, an ideal is a subset of transformations in our transformation semigroup which "absorbs" the elements that it comes into contact with. In other words, any time an element in our semigroup $\Sigma$ comes into contact with an element of the ideal, it becomes part of the ideal.

Formally, let $I$ be a nonempty subset of a semigroup $\Sigma$. $\Sigma I$ denotes the set $\{\sigma \circ i \mid \sigma \in \Sigma \text{ and } i \in I\}$. Likewise, $I\Sigma = \{i \circ \sigma \mid \sigma \in \Sigma \text{ and } i \in I\}$.

If $I$ is closed under left multiplication by any element in $\Sigma$, meaning $\Sigma I \subseteq I$, then we call $I$ a left ideal of $\Sigma$. If $I$ is closed under right multiplication by any element in $\Sigma$, meaning $I\Sigma \subseteq I$, then we call $I$ a right ideal of $\Sigma$. If $I$ is both a left and right ideal in $\Sigma$, meaning $\Sigma I \cup I\Sigma \subseteq I$, then $I$ is simply an ideal of $\Sigma$ (Cain 2013).

We define an ideal generated by an element as follows. Let $x \in \Sigma$ be arbitrary.

$L(x) = \Sigma^1 x = \{x\} \cup \Sigma x$ is the left ideal generated by x

$R(x) = x\Sigma^1 = \{x\} \cup x\Sigma$ is the right ideal generated by x

$$J(x) = \Sigma^1 x \Sigma^1 = \{x\} \cup \Sigma x \cup x\Sigma \cup \Sigma x\Sigma$$

is the principal ideal generated by x

Ideals generated by actions in planning problems can be thought of as sets of traces which contain that action and have a specific structure.

Suppose $a \in A$. The left ideal generated by $a$ is the set of all possible traces that end with $a$. The right ideal generated by $a$ is the set of all traces that start with $a$. The principal ideal generated by $a$ is the set of all traces which include $a$. We can further define ideals generated by sets of actions by taking the union of their individually generated ideals. We can also define ideals generated by traces, rather than a single action.

**Example: Generating an Ideal**   Let us implement and illustrate the ideals generated by an action in the "Drive" domain. Suppose we choose the transformation [ 7, 7, 4, 7, 7, 7, 7 ], which corresponds to drive-3-4 in our action set. The left ideal of drive-3-4 (the set of all possible traces that end with drive-3-4) is illustrated in Table 2.

If we attempt to append drive-3-4 to any sequence of actions other than {drive-1-3} or {begin, drive-1-3} we are

| Trace | Transformation |
|---|---|
| {**drive-3-4**} | $\begin{bmatrix} 1 & 2 & 3 & 4 & 5 & 6 & 7 \\ 7 & 7 & 4 & 7 & 7 & 7 & 7 \end{bmatrix}$ |
| {drive-1-3, **drive-3-4**} | $\begin{bmatrix} 1 & 2 & 3 & 4 & 5 & 6 & 7 \\ 4 & 7 & 7 & 7 & 7 & 7 & 7 \end{bmatrix}$ |
| {begin, drive-1-3, **drive-3-4**} | $\begin{bmatrix} 1 & 2 & 3 & 4 & 5 & 6 & 7 \\ 7 & 7 & 7 & 7 & 4 & 7 & 7 \end{bmatrix}$ |
| {..., **drive-3-4**} | $\begin{bmatrix} 1 & 2 & 3 & 4 & 5 & 6 & 7 \\ 7 & 7 & 7 & 7 & 7 & 7 & 7 \end{bmatrix}$ |

Table 2: The left ideal generated by "drive-3-4"

sent to the sink state, which corresponds to the transformation $[7, 7, 7, 7, 7, 7, 7]$ in the bottom row. This is because drive-3-4 is only applicable in state at-3. Similarly, the right ideal of drive-3-4 is the set { [ 7, 7, 4, 7, 7, 7, 7 ], [ 7, 7, 6, 7, 7, 7, 7 ], [ 7, 7, 7, 7, 7, 7, 7 ] }. We have {**drive-3-4**} itself, and the transformation corresponding to the trace {**drive-3-4**, end}. The two-sided, or principal, ideal includes the union of the left and right ideals, as well as the additional transformations [ 6, 7, 7, 7, 7, 7, 7 ] and [ 7, 7, 7, 7, 6, 7, 7 ] induced by the traces {drive-1-3, **drive-3-4**, end} and {begin, drive-1-3, **drive-3-4**, end} respectively.

## 4.2 Identifying Action Landmarks using Ideals

Let us now investigate how ideals can be used to identify action landmarks. Take, for example, the elevators domain previously defined and consider the grounded action *depart(f0, p0)*.

One way of using semigroup theory to determine whether this is a landmark action is by investigating the two-sided ideal of the "end" action. A benefit of using ideals, rather than inspecting the entire semigroup, is their size. Ideals are subsets of the semigroup; if we can identify an action which must occur in the plan (such as the "end" action), we can reduce the number of traces we inspect.

If there exists a transformation in this ideal that sends the start state to one of the goal states, we have a transformation that corresponds to a solution in our planning problem. More specifically, the factorization of this transformation corresponds to a sequence of actions that forms a valid plan. In this case, it would be even more beneficial to inspect the left ideal generated by "end", since we know it must be the final action in the solution. However, due to the limitations of Groups, Algorithms, Programming (GAP), a system for computational discrete algebra, we check the two-sided ideal instead.

Let us first implement the original planning problem and include all specified actions. We then check if there exists a transformation that sends the start state 9 to one of the goal states.

```
gap> Elements(I);
[ Transformation( [ 10, 14, 14, 10, 10, 14, 14, 10,
```

```
     14, 14, 14, 14, 14, 14 ] ),
  ...
  Transformation( [ 14, 14, 14, 14, 14, 14, 14, 14,
     10, 14, 14, 14, 14, 14 ] ),
  Transformation( [ 14, 14, 14, 14, 14, 14, 14, 14,
     11, 14, 14, 14, 14, 14 ] ),
  ... ) ]
```

Listing 1: Existence of solution before removing action depart(f0, p0) in the elevators domain

Since such a transformation exists (i.e, [ 14, 14, 14, 14, 14, 14, 14, 14, 10, 14, 14, 14, 14, 14 ] sends state 9 to state 10, which is a goal state), we conclude our problem is solvable. In order to identify *depart(f0, p0)* as a landmark action, we can remove it from our set of generators and repeat the previous method.

```
gap> Elements(I);
[ Transformation( [ 14, 14, 14, 14, 10, 11, 12, 13,
     14, 14, 14, 14, 14, 14 ] ),
  ...
  Transformation( [ 14, 14, 14, 14, 14, 13, 13, 14,
     14, 14, 14, 14, 14, 14 ] ),
  Transformation( [ 14, 14, 14, 14, 14, 14, 14, 14,
     14, 14, 14, 14, 14, 14 ] ) ]
```

Listing 2: Existence of solution after removing action depart(f0, p0) in the elevators domain

Inspecting the images of these transformations restricted to state 9, which corresponds to our start state, we see that their union is $\{14\}$, which corresponds to our sink state. Since there no longer exists a transformation from our start to a goal state, our problem has become unsolvable and we conclude that this is a landmark action. This process can be repeated iteratively to determine the complete set of action landmarks in a planning problem.

# 5  Dead Ends

A dead-end is a state from which it is impossible to reach the goal state by executing a sequence of actions (Ghallab, Nau, and Traverso 2016). Semigroups can be used to study dead ends in planning problems by analyzing the structure of the semigroup and identifying elements that correspond to transformations from which it is not possible to reach the goal state. The $\mathcal{L}$ relation, an equivalence relation which we define in the following section, partitions our semigroup into sets of transformations with the same image. Thus, if you have an action that sends you to a dead end, or set of dead ends, the $\mathcal{L}$-Class of that action will consist of traces which do the same.

Formally, an action $a$ will lead to a dead end if, in the ideal generated by $a$, our goal state is not in the union of the images of the transformations restricted to the start state. Essentially, this means no sequence of transformations which include $a$ and begin at the start state will result in our goal state, and thus $a$ is an action that leads to a set of dead end states. These correspond to transformations we can "throw away". Additionally, one can analyze sequences of actions, rather than a single action, and identify arbitrary traces that

result in dead ends.

One benefit of studying dead-ends with this technique is that it allows for more general reasoning, rather than just identification. For example, suppose that the ideal generated by an action does not include a transformation from the start state to the goal state. This implies that no solution to the given planning problem includes this action. However, if a goal state is in the image of some transformation in the ideal (not necessarily restricted to the start state), then there exist traces including this action in which our goal is still reachable. If we want a solution that includes this action, this implies that we may want to consider choosing a different start state. We provide an example of this in Section 5.2

## 5.1  Green's Relations

Green's relations are considered by many to be the most fundamental tool in understanding a semigroup (Howie 2002). They give information on the structure of a semigroup and how the elements interact based on the ideals they generate.

**The L, R, J, D, and H Relations**   Let $S$ be a set, $\Sigma^1$ be a transformation semigroup with adjoined identity, and suppose $x, y \in \Sigma^1$ are arbitrary transformations. We define the relations $\mathcal{L}$, $\mathcal{R}$, and $\mathcal{J}$ as follows:

$$x \mathcal{L} y \iff \Sigma^1 x = \Sigma^1 y$$
$$x \mathcal{R} y \iff x\Sigma^1 = y\Sigma^1$$
$$x \mathcal{J} y \iff \Sigma^1 x \Sigma^1 = \Sigma^1 y \Sigma^1$$

This implies $x \mathcal{L} y$ if they generate the same left ideal and $x \mathcal{R} y$ if they generate the same right ideal. Note that $x$ and $y$ could be a single action or a sequence of actions.

We define the kernel of a transformation $\alpha : S \to S$ as $ker(\alpha) = \{(x, y) \in S \times S \mid \alpha(x) = \alpha(y)\}$. That is, the kernel of $\alpha$ partitions $S$ into sets of elements having the same image under $\alpha$. The $\mathcal{L}$ and $\mathcal{R}$ relations are closely related to the kernel and image of a transformation:

$$x \mathcal{R} y \implies \ker x = \ker y$$
$$x \mathcal{L} y \implies \operatorname{Im} x = \operatorname{Im} y$$

The $\mathcal{R}$ relation partitions our semigroup into sets of transformations with the same kernel. From a planning perspective, this is equivalent to sequences of actions that are applicable to the same set of states. Likewise, the $\mathcal{L}$ relation partitions our semigroup into sets of transformations with the same image. From a planning perspective, this is equivalent to sequences of actions that send us to the same set of states. These relations are equivalence relations and the corresponding equivalence classes are called $\mathcal{L}$-classes, $\mathcal{R}$-classes, and $\mathcal{J}$-classes (Cain 2013).

By the definition of the composition of two binary relations, we have that

$$\mathcal{L} \circ \mathcal{R} = \{(x, y) \in \Sigma^1 \times \Sigma^1 : (\exists s \in \Sigma^1) \text{ such that } x \mathcal{R} s \mathcal{L} y\}$$

Note that $\mathcal{L}$ and $\mathcal{R}$ commute, that is, $\mathcal{L} \circ \mathcal{R} = \mathcal{R} \circ \mathcal{L}$. Thus, $\mathcal{L} \circ \mathcal{R}$ is the smallest equivalence relation $\mathcal{L} \vee \mathcal{R}$ containing both $\mathcal{L}$ and $\mathcal{R}$. The interested reader is referred to the proof of Lemma 2.1 in (Clifford and Preston 1964). Furthermore, it is a well known result that the intersection of two equivalence

| Trace | Transformation |
|---|---|
| {drive-1-3, drive-3-4} | $\begin{bmatrix} 1 & 2 & 3 & 4 & 5 & 6 & 7 \\ 4 & 7 & 7 & 7 & 7 & 7 & 7 \end{bmatrix}$ |
| {drive-2-4} | $\begin{bmatrix} 1 & 2 & 3 & 4 & 5 & 6 & 7 \\ 7 & 4 & 7 & 7 & 7 & 7 & 7 \end{bmatrix}$ |
| {drive-3-4} | $\begin{bmatrix} 1 & 2 & 3 & 4 & 5 & 6 & 7 \\ 7 & 7 & 4 & 7 & 7 & 7 & 7 \end{bmatrix}$ |
| {drive-4-3, drive-3-4} | $\begin{bmatrix} 1 & 2 & 3 & 4 & 5 & 6 & 7 \\ 7 & 7 & 7 & 4 & 7 & 7 & 7 \end{bmatrix}$ |

Table 3: The Green's $\mathcal{L}$-class of "drive-3-4"

| Trace | Transformation |
|---|---|
| {drive-3-1} | $\begin{bmatrix} 1 & 2 & 3 & 4 & 5 & 6 & 7 \\ 7 & 7 & 1 & 7 & 7 & 7 & 7 \end{bmatrix}$ |
| {drive-3-4, drive-4-2} | $\begin{bmatrix} 1 & 2 & 3 & 4 & 5 & 6 & 7 \\ 7 & 7 & 2 & 7 & 7 & 7 & 7 \end{bmatrix}$ |
| {drive-3-4, drive-4-3} | $\begin{bmatrix} 1 & 2 & 3 & 4 & 5 & 6 & 7 \\ 7 & 7 & 3 & 7 & 7 & 7 & 7 \end{bmatrix}$ |
| {drive-3-4} | $\begin{bmatrix} 1 & 2 & 3 & 4 & 5 & 6 & 7 \\ 7 & 7 & 4 & 7 & 7 & 7 & 7 \end{bmatrix}$ |

Table 4: The Green's $\mathcal{R}$-class of "drive-3-4"

relations is also an equivalence relation. Consequently, we have that the relations $\mathcal{L} \circ \mathcal{R}$ and $\mathcal{L} \cap \mathcal{R}$ are both equivalence relations.

Indeed, the aforementioned relations are widely studied and have their own names. We define $\mathcal{D}$ and $\mathcal{H}$ as $\mathcal{D} = \mathcal{L} \circ \mathcal{R}$ and $\mathcal{H} = \mathcal{L} \cap \mathcal{R}$. More formally:

$$x \mathcal{D} y \iff \exists s \in \Sigma^1 \text{ such that } (x \mathcal{R} s) \wedge (s \mathcal{L} y)$$
$$x \mathcal{H} y \iff (x \mathcal{R} y) \wedge (x \mathcal{L} y)$$

The $\mathcal{D}$ relation partitions our semigroup into sets of transformations having the same rank, where the rank is defined as the number of possible output values of the transformation (Dolinka and East 2015).

**Example: The Green's Relations of a Planning Problem**
Suppose we take the "Drive" example outlined above, add invertible drive actions, and once again choose the transformation corresponding to the action drive-3-4.

The $\mathcal{L}$-class of drive-3-4 consists of the transformations in Table 3. We observe that the image of all these transformations is the same. Thus, we have a set of traces that ensure we end in one of $\{4, 7\}$. Note that traces are determined by factoring the induced action in terms of its generators, and thus are not necessarily unique. Conversely, the $\mathcal{R}$-class of drive-3-4 consists of the transformations in Table 4. We observe that the kernel of all these transformations is the same, having the classes $\{3\}, \{1, 2, 4, 5, 6, 7\}$. Thus, we have a set

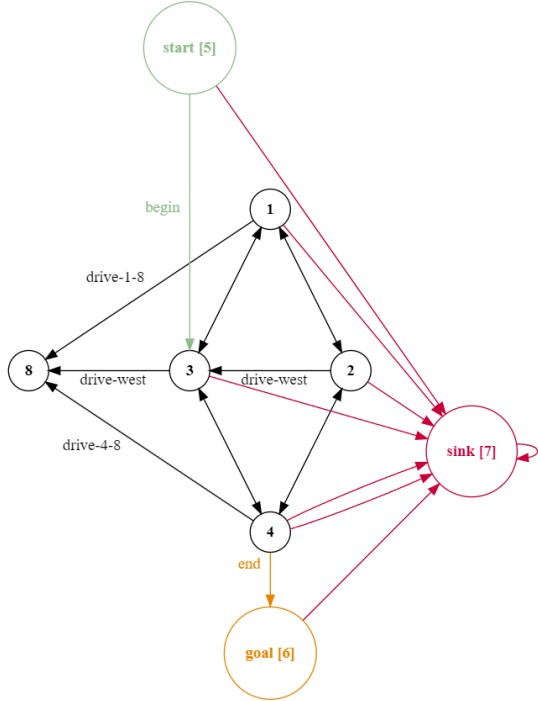

Figure 3: "Drive" state transition graph with a dead end

of traces that are applicable in state 3, and send the remaining class to the sink state.

## 5.2 Reasoning about Dead Ends using Green's Relations

Let us take the same drive domain with invertible actions used above, update our start state, and add an action drive-west. If we drive west from state 3, we end up at a new dead-end state which we call state 8. To make the structure of the problem more interesting, suppose we also add actions drive-1-8 and drive-4-8. The new state transition diagram is illustrated in Figure 3. We certainly want to avoid state 8, as it ensures we will never reach our goal. However, suppose we want to see the sunset, so we want to perform the action drive-west at some point. Let us use semigroup theory to drive west and still reach our goal.

We implement the planning problem, and then inspect the ideal of drive-1-8 using our previous methods to see if there exists a plan which includes this transformation and takes us to our goal state. Our goal state, which corresponds to state 6, is not in the image of any of these transformations. In fact, all sequences of actions that include drive-1-8 can only take us to states $8$ or $7$ (which is our sink state).

```
gap> Elements(I);
[ Transformation( [ 8, 7, 7, 7, 7, 7, 7, 7 ] ),
  ...
  Transformation( [ 7, 7, 7, 7, 7, 7, 7, 7 ] ) ]
```

Listing 3: Two-sided ideal of drive-1-4

Thus, we conclude that drive-1-8 will result in a dead end, no matter what our initial state is. The $\mathcal{L}$-class of drive-1-8 will consist of traces which behave in the same manner since they generate the same left ideal, and so we can freely eliminate any actions and traces present in that class because they also result in a dead end.

```
gap> Elements(GreensLClassOfElement( S,
    Transformation( [ 8, 7, 7, 7, 7, 7, 7, 7 ] ) )
    );
[ Transformation( [ 8, 7, 7, 7, 7, 7, 7, 7 ] )
 ...
 Transformation( [ 7, 7, 7, 8, 7, 7, 7, 7 ] ) ]
```

Listing 4: L-Class of drive-1-8

Inspecting the left ideal, we note the presence of and eliminate the action drive-4-8.

Now, let us consider the action drive-west.

```
gap> Elements(I);
[ Transformation( [ 1, 7, 7, 7, 7, 7, 7, 7 ] ),
 ...
 Transformation( [ 6, 7, 7, 7, 7, 7, 7, 7 ] ),
 Transformation( [ 7, 1, 7, 7, 7, 7, 7, 7 ] ),
 ...
 Transformation( [ 7, 7, 7, 7, 7, 7, 7, 7 ] ) ]
```

Listing 5: Two-sided ideal of drive-west

Inspecting the ideal of drive-west using our previous methods, we note that there is no transformation which takes us from state 3 to state 6. Thus, as it stands, our problem is unsolvable. However, we note that there exists a transformation which takes us from state 1 to state 6. This implies that if we choose state 1 instead of state 3 as our starting state, there exists a sequence of actions which includes drive-west and takes us to our goal.

## 6 Invertibility

Identifying invertible subsets of traces in planning is useful because it allows us to simplify the state space we consider.

### 6.1 $\mathcal{D}$-class Structure

Green's relations are useful in investigating both the local and global structure of a semigroup. We first note that in a finite semigroup, the $\mathcal{J}$ and $\mathcal{D}$ classes coincide. The interested reader is referred to a proof in (Pin 2010). We now introduce egg-box diagrams, which allow us to visualize $\mathcal{D}$-classes and how elements within a $\mathcal{D}$-class interact, thus telling us about the local structure of the semigroup. Additionally, the partial ordering of $\mathcal{D}$-classes tells us about the global structure of the semigroup.

### 6.2 Egg-box Diagrams

Recall $\mathcal{D} = \mathcal{L} \circ \mathcal{R}$, which implies every $\mathcal{D}$-class is a union of $\mathcal{L}$-classes and $\mathcal{R}$-classes. On the other hand, if an $\mathcal{L}$-class $L_x$ and $\mathcal{R}$-class $R_y$ intersect, then there exists some element $z \in L_x \cap R_y$. This implies $xLzRy$ and so $xDy$, and so $L_x$ and $R_y$ are both contained within the same $\mathcal{D}$-class. As a result, an $\mathcal{L}$-class and an $\mathcal{R}$-class only intersect when they both belong to the same $\mathcal{D}$-class (Cain 2013).

Thus, we can visualize $\mathcal{D}$-classes by using so called "egg-box diagrams". We arrange the elements of a $\mathcal{D}$-class in a grid, where each column is an $\mathcal{L}$-class and each row is an $\mathcal{R}$-class. Every cell in the grid is the intersection of the $\mathcal{L}$-class and $\mathcal{R}$-class containing that cell (Howie 1995).

The arrangement of $\mathcal{L}$-classes and $\mathcal{R}$-classes within an individual $\mathcal{D}$-class correspond to how elements relate to one another. Specifically, multiplying an element from the left side corresponds to moving down the column of the associated $\mathcal{L}$-class, and multiplying an element from the right side corresponds to moving to the right in the row of the associated $\mathcal{R}$-class.

If there exists an element $y \in \Sigma$ such that $xyx = x$, then we call $x$ a regular element. If a $\mathcal{D}$-class in a semigroup $\Sigma$ contains a regular element, then every element of $\mathcal{D}$ is regular. Regular $\mathcal{D}$-classes are of great interest in semigroup theory and have been studied extensively because they capture some notion of invertibility. In planning problems, they correspond to an equivalence class of invertible traces. The $\mathcal{D}$-classes of a semigroup are highly useful in locating the inverse of an element, in the sense that $x = xyx$ and $y = yxy$. In fact, if $\alpha$ is an element of a semigroup, then every inverse $\alpha^{-1}$ must lie in the same $\mathcal{D}$-class as $\alpha$ (Howie 1995).

The quasiorders which correspond to Green's relations induce a partial ordering of the corresponding classes (Pin 2010). As we "climb down" our lattice of $\mathcal{D}$-classes, the rank of our class goes down. From a planning perspective, this means that the number of possible states we can end up in by executing a given trace of actions goes down. In a finite monoid, invertible elements form the top $\mathcal{D}$-class, which is a group, and zero always forms a one-element bottom $\mathcal{D}$-class. Every semigroup has at most one minimal $\mathcal{D}$-class (Pin 2010).

In the context of planning problems, the $\mathcal{D}$-class of an element consists of all transformations which share either the same image or kernel. On the other hand, the $\mathcal{H}$-class of an element consists of all transformations which share exactly the same image and kernel. Thus, the traces which make up an $\mathcal{H}$-class are, in some sense, equivalent since each trace is applicable in the same set of states and will ensure we end in the same set of states. In fact, these transformations may form a group (Clifford and Preston 1964).

**Example: Locating Inverses**  "egg-box diagrams" allow one to easily identify the regular $\mathcal{D}$-classes of a semigroup. Let us illustrate the $\mathcal{D}$-class structure of the "Drive" domain with added invertible drive actions. By replacing traces with their corresponding transformations, one may more easily identify a regular $\mathcal{D}$-class, as seen in Figure 4. Note that the "-" character is used to represent the sink state. Each element in the regular $\mathcal{D}$-class has an inverse, which we can identify by looking at the element opposite the diagonal. For example, the inverse of the transformation [-, -, -, 2, -, -] (row 4, column 2) is [-, 4, -, -, -, -] (row 2, column 4). The elements along the diagonal, marked with an asterisk, are their own inverses.

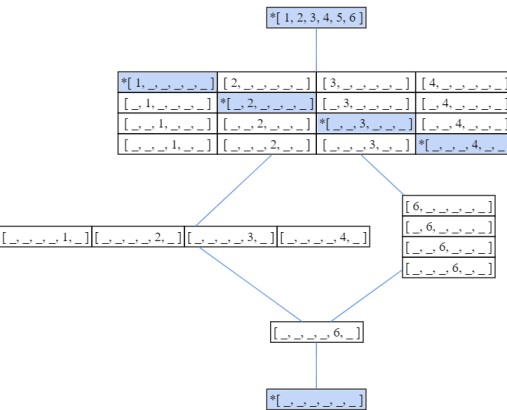

Figure 4: "Drive" $\mathcal{D}$-classes with corresponding transformations

## 7   Related Work

We begin with a survey of semigroup methods in computer science, starting with the question of why we look at computational models from an algebraic perspective in the first place. In *Finite Computational Structures and Implementations: Semigroups and Morphic Relations* (Egri-Nagy 2017), Dr. Attila Egri-Nagy suggests that generalizing existing models of computation to semigroup theory will help solve open problems in software and hardware engineering. In turn, the mathematical investigation relies on the tools of high-performance computing, forming a positive feedback loop between computer science and abstract algebra. Egri-Nagy argues that semigroups are the underlying mathematical structures of computers.

Semigroup theory has deep connections to automata theory and has been used to prove a variety of results. For example, see *Green's Relations and their Use in Automata Theory* (Colcombet 2011), in which Dr. Thomas Colcombet uses Green's relations to prove four classical results related to automata theory: the result of Schützenberger characterizing star-free languages, the theorem of factorization forests of Simon, the characterization of infinite words of decidable monadic theory due to Semenov, and the result of determinization of automata over infinite words of Mc-Naughton.

In an *Algebraic Analysis of Simple Computer Science Systems* (Morris et al. 2013), researchers apply semigroup theory to the 3-Queens Puzzle; a problem in computer science close to the nature of planning problems. Their work focuses on holonomy decomposition (which is a more efficient extension of the Krohn-Rhodes decomposition) and what insights it can provide. The Krohn-Rhodes decomposition theory is based on the idea that any finite-state automaton can be broken down into a series of "atomic" machines that depend on each other in a certain way. Since these atomic machines can't be broken down any further, they are like the prime factors in the process of breaking down an integer. There are two types of irreducible components: primes, whose semigroup of transformations is a simple group, and units, whose semigroup is contained in the identity-reset flip-flop with two states. Holonomy decomposition gives us the formal and cognitive tools to analyze and understand a finite deterministic automaton's global static and dynamic computational structure. More specifically, in their investigation of the 3-Queen's puzzle, they find that holonomy decomposition provides an interpretable hierarchical coordinate system.

## 8   Future Work

It remains to be determined how the invariants and equivalence classes induced by algebraic structures can provide a means of formulating equivalent problems through projection operations and classifications. Semigroups have been used in edge representation in machine vision models, and there exists a sound basis for recognition schemes that can be used in projection operations to obtain coarser or finer classifications depending on the parameters chosen (Hadingham 1990). Analogously, Green's relations have been used to break down problems in automata theory (Fleischer and Kufleitner 2019). However, a similar approach to classifying planning tasks remains an open and important problem.

Furthermore, one may investigate the use of hierarchical decompositions to exploit the hierarchical nature of planning problems. Information can only move in one direction along a partial order, so functional abstraction is possible. In a computational setting, semigroup theory has been used to decompose systems in the fields of biology, physics, psychology, philosophy, and games (Rhodes, Nehaniv, and Hirsch 2009; Egri-Nagy and Nehaniv 2008). Furthermore, understanding the algebraic structure underlying a model can inform algorithms for solving it. For example, by applying semigroup theory to the 3-Queens Puzzle, which can be modelled as a planning problem, holonomy decomposition can provide an interpretable hierarchical coordinate system (Morris et al. 2013). The idea of coarse-graining, which is throwing away information selectively to make models that are easier to understand, is yet another potential application to looking at these hierarchical breakdowns of planning problems.

Finally, there exist numerous algorithms for computing the structure of finite transformation semigroups (Linton et al. 2002). A number of approaches have been developed in computational semigroup theory in order to calculate fundamental properties of semigroups without enumerating all elements (East et al. 2019). These methods provide for local computations (concerning single equivalence classes) without computing the whole semigroup, as well as for computing the global structure of the semigroup. With respect to classical planning, such approaches can be used to compute the local structure of a single action, and the global structure of the planning problem and its solution space.

## 9   Conclusion

Algebraic modeling allows us to reveal insights into the hidden structures and symmetries of computational systems. Huge state transition tables are like quark-level descriptions for biological creatures (Egri-Nagy 2017). The semigroup

formalism offers a crucial framework for comprehending such systems, as it is necessary to recognize substructures and how they interact. Advantages of such an approach may include the potential for novel methods of task reformulation, analysis of traces and approaches to bisimulation, improvement of heuristics, and novel decompositions and factorizations.

In this work, we outline a rigorous algebraic formalism connecting planning problems and transformation semigroups. We demonstrate the correspondence between the algebraic model and the solution space of a planning problem. More specifically, we demonstrate the correspondence between elements of the transformation semigroup and traces induced by a planning problem. We compute some elementary examples with interesting properties and perform some visualizations to determine some basic properties of the semigroup induced by these planning problems, such as their Green's structure and inverses.

The goal of this work mainly lies in understanding, that is, establishing a sound, complete, and meaningful connection between the state transition models induced by planning problems and transformation semigroups. We are able to apply algebraic results to help understand the state space of a planning problem and study its solutions, and so we establish that such formal modeling is relevant and effective. We hope that future work will expand on the foundation established here, and explore further possibilities with deeper connections between semigroups and classical planning.

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
