# OpenReview forum: "From State Spaces to Semigroups: Leveraging Algebraic Formalism for Automated Planning"
_icaps-conference.org/ICAPS/2023/Workshop/HSDIP — ICAPS HSDIP 2023_

### Official Review · Reviewer_NLKp · 2023-04-14
**Interesting idea but poorly written. Theoretical paper that lacks precise mathematical formulations.**

**Rating:** 5
**Confidence:** 4

**Review:**

Short summary:

The authors advocate the use of algebraic semigroups as means to formally study classical planning tasks. They establish connections between several central classical planning notations -- reachability, landmarks, and dead ends -- to standard concepts in semigroup literature.

General comments:

Cross-connections between planning and formalism of other communities are of wide interest in the planning community. Thus the paper has the potential of yielding an interesting discussion at the workshop, although admittedly the topic is not really in line with the workshop's core subjects (heuristics and search). Overall, however, the paper suffers from a very poor and imprecise write-up even for workshop standards. I was not able to follow all details; the connections between standard semigroup notions and planning concepts are not worked out well. This is in so far critical has the scope of the paper is establishing exactly that, the theoretical connection between semigroups and classical planning. Moreover, it is not really clear why this connection may even be fruitful to look at at all. Defining the semigroup induced by a planning task entails constructing the entire state space in the first place, which per se makes it kind of pointless as a tool for solving the planning task (as advertised by the authors). Lastly, the related work section seems to focus on the wrong related work. All in all, I really like the idea of the paper. But, the realization of that idea leaves much room for improvement.

More detailed comments:

(1) The paper unfortunately lacks concise mathematical formulations, which makes it difficult to follow the discussion.

- This already starts with the definition of the semigroup that is induced by a planning task. Actually, it is still not entirely clear to me what that precise formal definition is exactly. Quoting the text: ``A transformation semigroup consists of a set of states Σ acted upon by a semigroup of transformations S''. How is one supposed to make sense of this? Transformations are not defined before this sentence; but even with that definition in mind, I am still not able to parse this definition. More specifically, a semigroup is composed of a set S over which a binary operation \circ is defined. Planning tasks are interpreted as transformation semigroup, where, presumably, S is a set of transformations, and \circ is a combination of those transformations. But how is the set S instantiated concretely (I presume the set of all action sequences)? What is \circ (I presume action sequence concatenation)? How exactly is the planning task's semantics represented?

- There are many notions that are used but never defined. Some standard algebraic notions, like monoids, neutral element, inverse element, I can still remember -- but they should be defined briefly nonetheless. There are also other, not so common ones, like regular element or adjoined identity that lack a proper definition. More critically, however, the paper also uses some non-standard notions/notation without definition, e.g., the second paragraph in the Idempotents section is essentially impossible to understand (what is T_x, what is Im; what is I?); in the definition of L(x)/R(x) what is S^1?; the multiplication of two sets (such as SI) has never been defined.

- The theory may also contain some bugs, so far as I can tell given that many concepts have not been defined. In particular, why should x^3=x for an idempotent element (defined as x^2 = x) imply that it is the inverse element of itself? First, note that it actually holds for all idempotent elements that x^3 = x^2 x = x x = x^2 = x. The inverse element x^{-1} is (typically) defined as x x^{x-1} = e. However, for idempotent element with x != e, it holds that x x = x; so if x^{-1} was x then x x^{-1} = x x = x contradicting the assumption x != e. The definition of L(x) and R(x) states that L(x) (R(x)) is the left (right) ideal generated by x. Doesn't this mean that S L(x) = L(x)? And if so, why should this be satisfied? (Symmetrically for R(x).)

(2) Relevance with respect to classical planning techniques / the workshop specifically.

- Also the description of the relation of the introduced semigroup concepts and standard concepts in classical planning is too abstract. What are the specific steps that one could take to extract landmarks from the semigroup formulation? What are the concrete steps to check reachability? What are the steps to extract dead end information?

- More importantly, in particular with respect to the workshop's topics, how is one supposed to make use of these semigroup concepts for devising practical planning algorithms? The authors mention abstraction. But if one needs to construct the state space to even define the semigroup, this kind of defeats the purpose of abstraction, as the construction solves the task in the first place. Similarly for reachability, dead ends, or heuristics. If the state space can be constructed, all of them can simply be solved by standard graph analysis methods. What does the semigroup interpretation offer over that, and why should it be discussed in a workshop on search and heuristics?

(3) Text structure. This is maybe only a minor point, but it kind of amplifies my first point. The text generally feels very unstructured. Basically, the core paper is just dumped into a single section. But many parts of this section don't seem to be directly related. Idempotents don't seem to be used anywhere, so it doesn't really make sense to have this as a subsection of Green's Relations. While Ideals establish the basis of the relations defined afterward, this is more something that belongs to preliminaries, and should neither be part of the Green's Relations section. Some of the paragraph also appear to be incorrectly placed. The discussion of landmarks and dead ends has nothing to do with Green's relations. Where are these relations used anyway?

(4) Related work. The authors focus mostly on discussing general semigroup literature. There however exist works exploring the connection of semigroups and automata theory / transition system analysis, which appear to be much closer to the topic of the paper. What are the findings of this paper that haven't been discussed in those prior works?

If the authors manage to improve the write-up wrt. (1) and (2), I think the paper could make a nice workshop addition. However, as it stands, the paper is not quite in shape for publication.

---

> ### Author Response · Authors · 2023-04-27
> **Response for reviewer NLKp**
>
> Dear Reviewer NLKp,
>
> Thank you for taking the time to review our paper and for providing valuable feedback. We greatly appreciate your insights and inquiries, and we believe that addressing the points you have raised will undoubtedly enhance the readability and clarity of our work.
>
> We apologize for any confusion caused by the lack of clear mathematical definitions in our paper. To address this, we will prioritize introducing all definitions used throughout the work and outlining the steps needed to instantiate a planning problem as a transformation semigroup. This will enhance the paper's readability and make the connections between standard semigroup notions and planning concepts clearer.
>
> Additionally, we appreciate your specific inquiries regarding the set S, action sequence concatenation (\circ), T_x, Im, I, S^1, and the inverse element x^{-1}. We will provide clear and concise definitions for these concepts in the revised version of the paper, making it easier for readers to understand and follow the work. For example, the image of a transformation f, denoted Im_f,  is the set of all the possible output values of that transformation. I is the identity transformation, which sends every element to itself. S^1 is a semigroup S with an identity transformation adjoined. These changes will address many points, including:
>
> > How is the set S instantiated concretely?
> The set S is the set of all action sequences, and \circ is action sequence concatenation. The set S is generated by taking the set of actions, and applying \circ to yield all possible action sequences.
>
> > What is T_x (Idempotents section)?
> T_x is a transformation semigroup (our apologies, using the variable S would have made this more clear). This second paragraph states that the transformation restricted to its image is the identity transformation with respect to this restricted set of states.
>
> > The inverse element x^{-1}
> In semigroup theory, an inverse of x is defined as an element x’ in S for which x = x x' x and x' x x' = x'. In this case, if we have that x is idempotent, we have x = x^3, and thus setting x’ = x satisfies this definition of an inverse.
>
> > Doesn't this mean that S L(x) = L(x)?
> This is correct, and satisfied by definition. Multiplying (applying \circ to) any element in L(x), by any element in S on the left will yield an element in L(x).
>
> Regarding relevance, we agree that the algebraic formalism of semigroup theory is a new perspective on studying the solution spaces of planning problems. We believe that exploring relations between sequences of actions using algebraic techniques is a potentially fruitful approach to this study, and hope that presenting the work at HSDIP will spark several ideas throughout the community.
>
> The semigroup interpretation of planning problems allows for more general analysis, as seen in our example on reachability analysis, where semigroup theory can identify more optimal start states. Our argument may come across as abstract, and we can address this in part by adding a concrete example of this in the paper. But elucidating these connections is part of our motivation for bringing the work to HSDIP. We further acknowledge that the potential connections to landmark extraction and reachability analysis can be made less abstract by explicitly outlining the steps taken in the corresponding examples.
>
> We believe that submitting the work is an opportunity to facilitate a discussion with the HSDIP community on what further connections to heuristics could look like. We hope to identify concrete applications in the planning community focused specifically on heuristics that we can further explore – we do not view scalability as a substantial concern, given that so many heuristic ideas involve projecting transition systems down to smaller state spaces that can be explicitly analyzed. Furthermore, we hope to eventually leverage existing algorithms in computational semigroup theory to calculate fundamental properties of semigroups without enumerating all elements.
>
> To improve clarity, we will focus more on concrete examples related to heuristics and search, introducing relevant concepts from semigroup theory only as they relate to said concepts. For example, we agree that in discussing landmark heuristics, we need only introduce ideals. We believe that this approach will make the work more accessible to readers and aid in enhancing its relevance. Lastly, we appreciate your suggestions regarding the related work section. We will focus more on work related to semigroups and automata theory / transition system analysis, as suggested.
>
> Once again, thank you for your valuable feedback. It is an extremely thorough review for a workshop submission, and we very much appreciate all of the suggestions. We hope that you still see merit in the ideas to the point of making this a worthwhile discussion to continue at HSDIP, as we believe it would be a fascinating conversation to have with the HSDIP community more broadly.

---

### Official Review · Reviewer_imkc · 2023-04-25
**Interesting formalizations, some hints at analysis, would benefit from more clarity**

**Rating:** 5
**Confidence:** 4

**Review:**

Summary:

This paper introduces formalisms for framing planning problems in terms of algebraic semigroups, pointing out that semigroups could be used to analyze the solutions of planning problems. The authors point to some areas of interest for such analysis, including landmarks, dead ends, and categorizing actions according to Greens' relations.

Specific Feedback:

The paper begins a discussion of semigroups for planning problems in a thought-provoking way, but many of the concepts could be more clearly described if their planning-specific meanings were given up front and with more emphasis on how they can be used. Some of this could be done by restructuring, though some may require more information being added. It would greatly increase comprehensibility of the work if there was an early description (either early in the paper or early in each section) of which features of a planning problem will be represented using which groups, which structures can be analyzed using which techniques, and what benefits may be derived from identifying or utilizing each of these.

The discussion of landmarks seems to hint at interesting ideas and applications of these formalisms, but does not directly get to these applications. It would be very interesting to see a concrete example of using semigroup analysis to discover landmarks for a specific problem. Likewise, this section

It is mentioned in the introduction that semigroup analysis will only be feasible for small problem instances, and that the results on these small instances can be generalized to larger instances. This is not, however, followed up with such general results from analysis of small problems later in the paper. Some examples of extending the small-scale analysis to discoveries which would be useful in larger sized problems would help demonstrate the importance of this work.

This paper has the potential to provide a grounding for using semigroup analysis to understand the structure of planning problems. However, to do so it needs to be clearer in its presentation and provide more specifics on how this analysis can benefit heuristic search planning.

Minor points:

Figure 3 is hard to read, due to its vertical length and how small some components get when viewing the whole diagram.

The notation for I_m_eps and I_I_m_eps in the Idempotents section was not introduced before or explained after, that I could find.

D-classes are mentioned under the Landmarks section in a way that does not indicate what they are, or that they will be explained later.

Including the intersection and union symbols in Figure 4 would make the connections less ambiguous.

---

> ### Author Response · Authors · 2023-04-27
> **Response for reviewer imkc**
>
> Dear Reviewer imkc,
>
> Thank you for taking the time to review our paper. We appreciate your expertise and insights, and have taken your comments into consideration.
>
> We understand your concerns regarding the clarity of our presentation and the need for more specifics on how the analysis of semigroups can benefit heuristic search. Our driving motivation to bring this work to HSDIP in particular, is to facilitate exactly this sort of discussion with the community. We will revise our paper to provide more emphasis on the planning-specific meanings of the concepts introduced and how they can be used to analyze planning problems, and welcome any feedback you might have on these fascinating connections.
>
> To improve overall clarity, we can place more emphasis on a set of precise definitions used regularly throughout the work (common in the mathematics community, but rarely seen in planning literature), and additionally detail the steps required to instantiate a planning problem as a transformation semigroup. This will hopefully make the paper more approachable for the planning community and make clear how standard semigroup notions relate to planning concepts. We can also add an early description of which features of a planning problem can be represented (namely, the set of all possible sequences of actions), which structures can be analyzed using which techniques (i.e, landmarks can be analyzed using ideals), and what benefits may be derived from identifying or utilizing each of these.
>
> Regarding your comment on the discussion of landmarks, we agree that more concrete examples of using semigroup analysis to discover landmarks for specific problems would be helpful. We include an example of identifying a landmark in the elevators domain, however we may extend this example to analyze all actions. We further acknowledge that the potential connections to landmark extraction and reachability analysis can be made less abstract by explicitly outlining the steps taken in the corresponding examples. However, we believe that by submitting the work to HSDIP, we have the opportunity to start a deeper conversation with the community about potential future heuristic connections.
>
> Regarding the criticism of scalability of the work, we would like to highlight that our primary objective of bringing this to HSDIP also informs the potential application of these ideas in a scalable manner: abstraction-based heuristics maintain shortest path information on entire (yet abstract) state spaces. We hope to find similar connections and analysis on the various forms of abstractions known well in the planning community. I.e., we would like to bring the advanced methods from semigroup theory and algebraic analysis to spark discussion at HSDIP.
>
> We also appreciate your minor points, and will make changes to the paper to make Figure 3 clearer and to introduce and explain notation and concepts more effectively.
>
> Thank you again for your review and feedback. We hope that our revisions will address your concerns and improve the clarity and comprehensibility of our paper. We would love to have the opportunity to continue the discussion both here and in-person at HSDIP.

---

### Decision · Program_Chairs · 2023-05-06

**Decision:**

Accept

**Comment:**

We are happy to announce that the paper is accepted to the workshop. We agree with the reviewers that this provides an interesting new direction for examining the structure of planning problems.

For the final version, we ask that you address the concerns and corrections mentioned by the reviewers, especially around clarity and concreteness.